# Upregulation of Insulin-like Growth Factor-I in Response to Chemotherapy in Children with Acute Lymphoblastic Leukemia

**DOI:** 10.3390/ijms25179582

**Published:** 2024-09-04

**Authors:** Helin Berna Kocadag, Sarah Weischendorff, Silvia De Pietri, Claus Henrik Nielsen, Mathias Rathe, Bodil Als-Nielsen, Henrik Hasle, Anders Juul, Klaus Müller, Maria Ebbesen Sørum

**Affiliations:** 1Department of Pediatrics and Adolescent Medicine, Copenhagen University Hospital—Rigshospitalet, 2100 Copenhagen, Denmark; helin.berna.kocadag.01@regionh.dk (H.B.K.);; 2Institute for Inflammation Research, Copenhagen University Hospital—Rigshospitalet, 2100 Copenhagen, Denmark; 3Hans Christian Andersen Children’s Hospital, Odense University Hospital, 5000 Odense, Denmark; 4Department of Pediatrics, Aarhus University Hospital, 8200 Aarhus, Denmark; 5Department of Growth and Reproduction, Copenhagen University Hospital—Rigshospitalet, 2100 Copenhagen, Denmark; 6Department of Clinical Medicine, University of Copenhagen, 2200 Copenhagen, Denmark; 7International Research Centre for Endocrine Disruption of Male Reproduction and Child Health (EDMaRC), Copenhagen University Hospital—Rigshospitalet, University of Copenhagen, 2200 Copenhagen, Denmark

**Keywords:** acute lymphoblastic leukemia, IGF-I, growth factors, inflammation, citrulline, tissue damage, gut barrier

## Abstract

The treatment of childhood cancer is challenged by toxic side effects mainly due to chemotherapy-induced organ damage and infections, which are accompanied by severe systemic inflammation. Insulin-like growth factor I (IGF-I) is a key regulating factor in tissue repair. This study investigated associations between the circulating IGF-I levels and chemotherapy-related toxicity in pediatric acute lymphoblastic leukemia (ALL). In this prospective study, we included 114 patients (age: 1–17 years) with newly diagnosed ALL treated according to The Nordic Society of Paediatric Haematology and Oncology (NOPHO) ALL2008 protocol between 2013 and 2018. The patients’ plasma levels of IGF-I, and the primary binding protein, IGFBP-3, were measured weekly during the first six weeks of treatment, including the induction therapy. The patients’ systemic inflammation was monitored by their C-reactive protein (CRP) and interleukin (IL)-6 levels and their intestinal epithelial damage by their plasma citrulline levels. IGF-I and IGFBP-3 were converted into sex-and age-adjusted standard deviation scores (SDS) using 1621 healthy children as reference. At ALL diagnosis, IGF-I levels were decreased (median (quartiles): −1.2 SDS (−1.9 to −0.5), *p* = 0.001), but increased significantly following the initiation of chemotherapy, peaking on day 8 (0.0 SDS (from −0.8 to 0.7), *p* < 0.001). This increase correlated with the levels of CRP (rho = 0.37, *p* < 0.001) and IL-6 (rho = 0.39, *p* = 0.03) on day 15, when these markers reached maximum levels. A larger IGF-I increase from day 1 to 15 correlated with a slower recovery rate of the intestinal damage marker citrulline from day 15 to 29 (rho = −0.28, *p* = 0.01). Likewise, IGFBP-3 was reduced at diagnosis, followed by an increase after treatment initiation, and was highly correlated with same-day IGF-I levels. This study demonstrates a chemotherapy-induced increase in IGF-I, with a response that appears to reflect the severity of tissue damage and systemic inflammation, preceding CRP and IL-6 increases. IGF-I may have potential as an early reactive biomarker for acute toxicity in patients with ALL.

## 1. Introduction

The survival rates for childhood acute lymphoblastic leukemia (ALL) have markedly improved during the recent decades, now approaching 90% [1]. Despite this progress, the treatment of ALL still faces substantial challenges due to severe treatment-related toxicities [2,3,4]. Patients undergoing treatment for ALL are at risk of invasive, potentially life-threatening infections due to intestinal barrier damage and neutropenia induced by the highly intensive chemotherapy. This leads to prolonged hospitalization, extended antibiotic treatment, and, consequently, delays in subsequent antineoplastic treatment, potentially compromising survival [5,6,7]. There is substantial evidence suggesting that chemotherapy-induced damage to the intestinal epithelial barrier is a key initiating step in these toxicities due to the translocation of bacteria into the bloodstream [8,9]. Furthermore, even in the absence of positive blood cultures, bacterial products may cause non-specific systemic inflammation [10,11,12,13,14], which is thought to propagate organ damage, as demonstrated in the setting of hematopoietic stem cell transplantation (HSCT) [15,16,17,18,19]. The extent of chemotherapy-induced toxic reactions varies significantly among patients, and the underlying mechanisms for this interpersonal variability remain poorly understood. However, previous studies from our group and others have pointed to factors controlling tissue healing, such as growth factors, as potential contributors to elucidating this variability [20,21,22,23,24].

Insulin-like growth factor-I (IGF-I) is a peptide hormone mainly produced by the liver under the control of pulsatile growth hormone secretion [25]. Circulating IGF-I is primarily liver-derived and exerts endocrine actions. Still, other sources of IGF-I include intestinal smooth muscle cells and the intestinal epithelium, which are important for the auto- and paracrine actions of IGF-I [26,27,28]. IGF-I plays a crucial role in promoting the growth of almost all tissues [25], and has, in addition, anti-inflammatory effects [29,30,31]. In line with this, rodent studies have demonstrated that treatment with IGF-I promotes the regeneration of damaged epithelia and mitigates chemotherapy-induced mucositis [22,32,33]. Furthermore, plasma IGF-I levels have shown predictive value for the severity of systemic inflammation and liver sinusoidal obstruction syndrome in patients undergoing HSCT [21,24].

These findings suggest that inflammation, IGF-I production, and intestinal epithelial healing are functionally interrelated. In the present study of 114 pediatric patients with ALL, we demonstrate a chemotherapy-induced increase in IGF-I, which correlates with the severity of tissue damage and systemic inflammation. This suggests that IGF-I could potentially serve as an early reactive biomarker for acute toxicity in patients with ALL.

## 2. Results

A total of 114 pediatric patients with a median age of 4.2 (range: 1–17 years) diagnosed with ALL between 2013 and 2018 were included in this study. There was a slight predominance of the male sex (59%) over the female. The majority of patients had pre-B-cell ALL (87%) and 82% were classified as non-high risk at induction start. The patient characteristics are outlined in detail in Table 1.

### 2.1. IGF-I and IGFBP3

At the time point of diagnosis, before the initiation of treatment (day 1), median sex- and age-adjusted standard deviation scores (SDS) of IGF-I and IGFBP-3 were reduced compared to healthy controls (i.e., 0 SDS) (−1.2 SDS (interquartile range: −1.89 to −0.53), *p* = 0.001 and −1.9 SDS (−2.79 to −1.01 SDS, *p* < 0.001, respectively) (Figure 1). However, from day 8, we observed a significant increase in IGF-I (Figure 1A) and IGFBP-3 (Figure 1B) levels, which was sustained during the entire induction treatment (until day 29) (all *p* < 0.001). Subsequently, in the post-induction samples that were available (*n* = 53), we observed a significant decline in IGF-I levels from the end of induction treatment (day 29) towards day 43 (median: 0.48 vs. −0.52 SDS, *p* < 0.001), and towards day 36 for the IGFBP-3 levels (−0.38 vs. −1.90 SDS, *p* < 0.001) (Figure 1).

The IGF-I and IGFBP-3 levels were not associated with patients’ ALL immunophenotype or risk classification groups at any time point (all *p* > 0.05).

### 2.2. Systemic Inflammation and IGF-I

Due to a strong correlation between the IGF-I and IGFBP-3 levels at all measured time points (rho = 0.66–0.80, all *p* < 0.001), only the results for IGF-I are reported in the result Section 2.2 and Section 2.3.

Following a modest elevation at the time of ALL diagnosis, the CRP levels declined to a nadir around day 8, followed by a subsequent increase, peaking between day 13 and 17 (median: 25 pg/mL (range: 1–394)) (Figure 2A).

We investigated whether the IGF-I levels were associated with the CRP levels by stratifying the patients into two groups according to their median IGF-I level at day 8. Overall, the group of patients with high IGF-I on day 8 had higher CRP levels from day 12 to day 19 compared with the patients with low IGF-I on day 8, reaching statistical significance on day 15 (mean: 185% higher CRP, 95% confidence interval (CI): 28–634%, *p* = 0.01) and day 16 (mean: 160% higher CRP, 95%CI: 4–651%, *p* = 0.04) (Figure 2A). Similarly, we found a positive correlation between the relative increase in IGF-I levels from day 1 to day 8 and CRP levels on day 15 (rho = 0.37, *p* < 0.001) (Figure 2B).

A similar pattern was observed when examining the IL-6 levels, which peaked at day 15 (Figure 3A). A larger increase in IGF-I from day 1 to 8 correlated significantly with the IL-6 levels on days 15 (rho = 0.39, *p* = 0.03) and 22 (rho = 0.35, *p* = 0.05) (Figure 3B).

These findings collectively demonstrate a consistent positive correlation between the initial increase in IGF-I and the subsequent systemic inflammatory response.

### 2.3. Intestinal Mucosal Toxicity and IGF-I

During the induction treatment for ALL, patients experience varying degrees of gastrointestinal toxicity, characterized by nausea, vomiting, abdominal pain, and diarrhea, which is indicative of intestinal mucositis (IM), typically peaking around two weeks after initiating chemotherapy. To explore the relationship between IGF-I levels and intestinal toxicity, we assessed patients’ citrulline levels, as citrulline is selectively produced by viable enterocytes and recognized as a validated quantifiable marker of intestinal mucositis [35]. The citrulline levels decreased significantly after the start of chemotherapy, reaching nadir on day 15, coinciding with the peak of systemic inflammation, as indicated by the CRP levels, and then increased towards pre-chemotherapy levels on day 29 (Figure 4A), as previously observed in a subgroup of this study cohort [36].

We found no significant associations between the IGF-I levels and the decrease in citrulline from day 1 to nadir levels (day 15). However, we found a negative correlation between the increase in IGF-I from day 1 to day 15 and the increase in the citrulline levels from day 15 to day 29 (rho = −0.3, *p* = 0.01) (Figure 4B). This suggests that a larger increase in IGF-I is associated with a slower recovery rate of citrulline. Similar findings were observed regarding IGFBP-3 increase and the recovery rate of citrulline.

Next, we studied the degree of intestinal mucositis (IM) by clinical scoring for abdominal pain and diarrhea (grade 0 (0–1 points), grade I (2–3 points), and grade II (≥4 points)) and its association with IGF-I levels. The incidence of IM increased from day 1, peaking on day 15 (*p* < 0.001), where 32% of the patients had either grade I (24.5%) or grade II (7.3%) IM (Figure 5A). This was followed by a gradual decrease from day 15 to day 29 (*p* = 0.05). The IGF-I levels on day 8 were significantly higher in patients developing grade II IM on day 15 compared with patients with grade 0-I IM (median −0.22 vs. 0.88 SDS, *p* = 0.02) (Figure 5B).

## 3. Discussion

The present study investigated patients’ IGF-I and IGFBP-3 levels during the first six weeks of ALL therapy. We observed a rapid increase in the plasma levels of IGF-I and IGFBP-3 after the initiation of chemotherapy treatment, correlating with the severity of tissue toxicity in terms of an increased systemic inflammatory response and a delayed recovery of intestinal epithelial cells.

The finding of overall reduced IGF-I levels before the initiation of chemotherapy is in line with previous findings of reduced IGF-I levels in children with ALL at the time of diagnosis [37,38,39] and in children referred for HSCT [21,23,37]. Although our data do not explain the mechanism underlying reduced IGF-I at the ALL diagnosis time point, various factors known to affect IGF-I levels may be involved, including co-morbidities such as critical illness, hepatic diseases, and malnutrition [25,39,40,41,42,43].

To our knowledge, an increase in IGF-I levels in the first few weeks after initiation of chemotherapy treatment has not been described previously in patients treated with first-line chemotherapy. However, we have observed similar findings with a temporary increase in IGF-I levels within the first weeks after the initiation of high-dose chemotherapy in both pediatric and adult patients undergoing HSCT [21,24]. Other forms of acute stress, such as septic shock, have also been associated with an increase in IGF-I and IGFBP-3 levels within the first few days of hospital admission [44], and heavy exercise has been found to momentarily upregulate IGF-I production, both from the liver and other tissues [45]. Together with our study, these findings indicate that IGF-I may act as an acute phase reactant, known as a group of proteins and peptides that are released in response to various traumas, reflecting the severity of tissue damage. Importantly, the IGF-I response appeared to peak one week earlier than the well-established acute-phase reactant CRP.

The mechanism behind this IGF-I increase is at present unclear but may be due to a stress-induced release of GH, which stimulates IGF-I synthesis in the liver [46]. Although the regulation of locally produced IGF-I is not fully understood, another growth factor, glucagon-like peptide-2, increases in response to intestinal mucosal damage [47] and has been suggested to stimulate intestinal IGF-I production [26]. Indeed, our finding of a positive association between the rapid IGF-I increase and subsequent CRP levels points to a potential role of IGF-I as an early biomarker of systemic inflammatory reactions.

According to the existing evidence, IGF-I appears to exhibit both anti-inflammatory and epithelial protective properties [31,48,49]. IGF-I can reduce the ratio between pro- and anti-inflammatory cytokines [29], while also stimulating enterocyte proliferation and facilitating the regeneration of damaged epithelia, including the gut mucosa, as demonstrated in rodent models [22,32,33,50]. Our studies suggest that IGF-I is rapidly released upon chemotherapy initiation, with a response that may reflect the severity of the initiating mucosal damage, which starts out the cascade of reactions that leads to intestinal mucositis and subsequent systemic inflammation according to the model by Sonis [8]. This pattern parallels observations seen in several other protective biological systems, including cytokine antagonists, where elevated levels reflect more severe inflammation despite their anti-inflammatory properties [51,52,53,54]. Accordingly, IGF-I may be secreted as a protective factor, although the increase may be insufficient to effectively counteract the toxic effect of the chemotherapy. It can be speculated that supporting endogenous IGF-I production—such as through an optimized nutritional status or treatment with IGF-I/IGFBP-3—might be beneficial. However, the safety of exogenous IGF-I in this neoplastic setting must be carefully evaluated, given its mitogenic effects [55].

In a broader perspective, this study suggests that IGF-I should be considered as a dynamic hormone that not only influences tissue healing and inflammation but is itself also affected by tissue damage and acute inflammation. This is particularly important when evaluating a patient’s IGF-I status, as caution may be needed when interpreting a single plasma measurement.

The present study features a robust design with preprogrammed and consistent sampling across the entire induction period. It was conducted within a sizable, population-based cohort of similarly treated pediatric patients with ALL.

Our study was limited by the relatively small cohort, which resulted in a lack of the power to investigate associations between IGF-I and additional clinical outcomes. Furthermore, the clinical scoring of mucositis lacks specificity due to other potential causes of symptoms and the subjectivity of patient reporting. However, this was partially compensated for by the use of citrulline as an additional, more specific marker of gut damage. Additional studies in larger cohorts are required to further explore the potential clinical utilization of IGF-I as a predictive biomarker for treatment-related toxicities in ALL treatment.

## 4. Materials and Methods

### 4.1. Study Population

We prospectively included children (inclusion criteria: age 1–18 years) with newly diagnosed ALL undergoing treatment according to the Nordic Society of Pediatric Hematology and Oncology (NOPHO) ALL 2008 protocol at Rigshospitalet, University Hospital of Copenhagen, H.C. Andersen Children’s Hospital, Odense University Hospital and Aarhus University Hospital, Denmark, between March 2013 and December 2018.

The NOPHO ALL-2008 induction treatment lasted 29 days and includes intravenous (i.v.) vincristine at a 2 mg/m^2^ dose (max 2 mg) administered on days 1, 8, 15, 22, and 29, and i.v. doxorubicin at 40 mg/m^2^ on days 1 and 22. Intrathecal (i.t.) methotrexate was administered on days 1, 8, 15, and 29 [34]. In addition, oral corticosteroids were given daily, with prednisolone (60 mg/m^2^/day) administered from days 1 to 29 in the non-high-risk (non-HR) treatment group, and dexamethasone (10 mg/m^2^/day) from days 1 to 21 in the high-risk (HR) group [34]. Post-induction consolidation therapy was initiated on day 36, or when hematological laboratory parameters allowed. Patients assigned to standard risk, intermediate risk, and high-risk groups based on day-29 risk group assignment received high-dose methotrexate (a 24 h continuous infusion at a total dose of 5000 mg/m^2^) with intrathecal methotrexate, oral 6-mercaptopurine (25 mg/m^2^), and intramuscular pegylated asparaginase (1000 IU/m^2^) during the first week of consolidation therapy. The first week of consolidation therapy for patients in the high-risk group consisted of cyclophosphamide 440 mg/m^2^ i.v. for 5 days, etoposide 100 mg/m^2^ i.v. for 5 days, and intramuscular pegylated asparaginase (1000 IU/m^2^) as previously described in detail [34].

### 4.2. IGF-I and IGFBP-3

Plasma from ethylenediaminetetraacetic acid (EDTA) anti-coagulated blood was collected on treatment days 1, 8, 15, 22, and 29 (median 5, range 0–5 samples per patient). For 53 (47%) of the patients, additional plasma samples on days 36 and 43 were available. Before quantification, IGF-I was dissociated from its binding proteins. IGF-I and IGFBP-3 were quantified using IDS-iSYS IGF-I and IDS-iSYS IGFBP-3 assays (Immunodiagnostic Systems LTD, Bolton, UK) based on chemiluminescence technology. These assays have undergone validation and accreditation [21]. The inter-assay and intra-assay coefficients of variation were 7.2% and 2.1%, respectively. The lower limit of detection was 10 μg/L.

IGF-I and IGFBP-3 plasma concentrations were converted to sex- and age-adjusted SD-scores using a generalized additive model for location, scale, and shape [56]. Normal ranges have been previously established in a cohort of 1621 healthy children (891 males) aged from 1 to 18 years [57].

### 4.3. C-Reactive Protein, Interleukin-6, Citrulline and Intestinal Mucositis

During the four-week induction period, plasma CRP was measured on a median of 19 out of 29 days (range 4–29) for each patient as part of the clinical routine, using an automated immunoturbidimetric assay utilizing the Modular P module (Roche, Basel, Switzerland). When multiple measurements were taken per day, the highest level of CRP was used. Levels of interleukin-6 (IL-6) were measured in a subgroup of patients (*n* = 49) on days 8, 15, 22, and 29 using the Bio-Plex Pro Human Chemokine Assay (Bio-Rad, Hercules, CA, USA) on the Luminex platform (Luminex Corporation, Austin, TX, USA), according to the manufacturer’s instructions.

Plasma citrulline levels were measured on days 1, 8, 15, 22, and 29 as a marker of intestinal epithelial damage. These measurements were performed using a Waters Acquity Ultra-Performance Liquid Chromatography system coupled to a Sciex Qtrap 6500+ mass spectrometer (AB Sciex, Framingham, MA, USA) with L-citrulline (4,4,5,5-D4) (Cambridge Isotope Laboratories, MA, USA) as the internal standard [58]. 

Intestinal mucositis (IM) was scored on treatment days 1, 8, 15, 22, and 29 by a trained study nurse by summation of the scores for abdominal pain and diarrhea according to the National Cancer Institute Common Terminology Criteria for Adverse Events (NCI-CTACAE) 4.0 criteria. The scoring system was as follows: grade 0 (0–1 points), grade I (2–3 points), and grade II (≥4 points) [59].

### 4.4. Statistics

Changes in IGF-I and IGFBP-3 levels over time and associations with patient-specific characteristics (ALL phenotype, risk group, and treatment centre) were analysed using mixed models with an unstructured variance–covariance matrix with log-transformed IGF-I and IGFBP-3 levels (due to their skewness) and expressed as mean with associated 95% CI. Likewise, a mixed model with an unstructured variance–covariance matrix was used when investigating the association between dichotomized IGF-I level on day 8 and CRP over time. The Mann–Whitney U-test was used for all additional comparisons between groups. Correlation analyses were performed using Spearman’s rank-order correlation analysis for continuous outcomes. *p*-values < 0.05 were considered statistically significant. All statistical analyses were performed using R statistical software (Version 3.4.0, R Foundation for Statistical Computing, Vienna, Austria) [60].

## 5. Conclusions

In conclusion, this study reveals a significant pattern wherein the IGF-I levels are reduced in children with newly diagnosed ALL but increase in response to chemotherapy, with the intensity of this response reflecting the severity of toxicity. The finding of increased IGF-I levels preceding the increases in established inflammatory markers suggests its potential as an early biomarker for chemotherapy-induced tissue injury and inflammation. Our findings indicate that IGF-1 should be considered as a dynamic factor that increases in response to tissue damage, with implications potentially extending beyond the field of pediatric oncology.

## Figures and Tables

**Figure 1 ijms-25-09582-f001:**
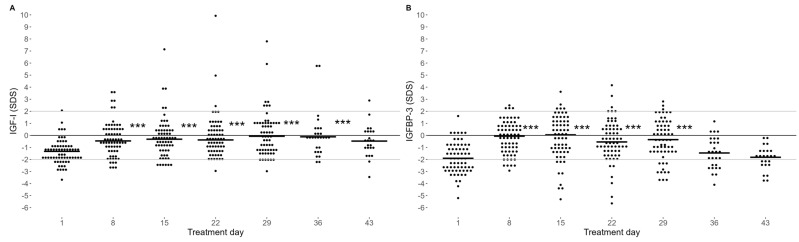
IGF-I (**A**) and IGFBP-3 (**B**) plasma levels as sex- and age-adjusted SD-scores. Day 1 refers to the start of induction treatment. Horizontal short line: median IGF-I and IGFBP-3. Consistent lines: mean (black) and reference interval (grey) of IGF-I for healthy children. Statistical evaluation using a mixed-model analysis indicates that IGF-I levels from day 8 to 36 and IGFBP-3 levels from day 8 to 29 were significantly higher than on day 1, *** *p* < 0.001. Number of samples: (**A**) day 1 (*n* = 84), day 8 (*n* = 87), day 15 (*n* = 84), day 22 (*n* = 84), day 29 (*n* = 82), day 36 (*n* = 36), day 43 (*n* = 31). (**B**) Day 1 (*n* = 65), day 8 (*n* = 68), day 15 (*n* = 64), day 22 (*n* = 65), day 29 (*n* = 62), day 15 (*n* = 28), day 43 (*n* = 24).

**Figure 2 ijms-25-09582-f002:**
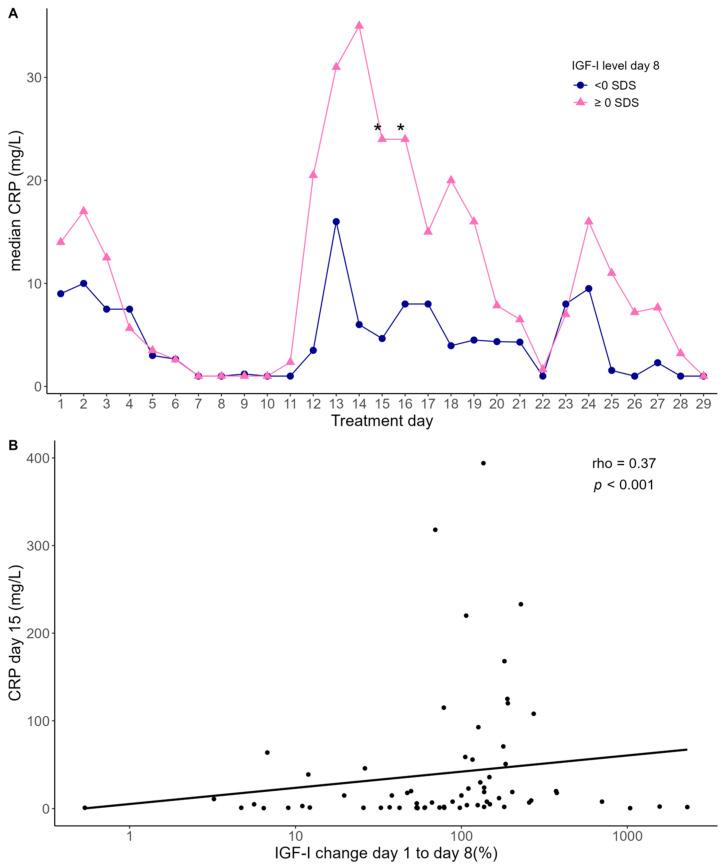
(**A**) Median CRP levels in two groups stratified according to the median level of IGF-I on day 8. Statistical evaluation using mixed-model analysis. Asterix indicates statistical differences in CRP levels between the two groups, day 15 (* *p* = 0.01) and day 16 (* *p* = 0.04). (**B**) Association between CRP levels on day 15 and relative change in IGF-I levels from day 1 to 8. Statistics by Spearman correlation analysis (rho = 0.37, *p* < 0.001). The black line represents a linear regression line fitted to the data.

**Figure 3 ijms-25-09582-f003:**
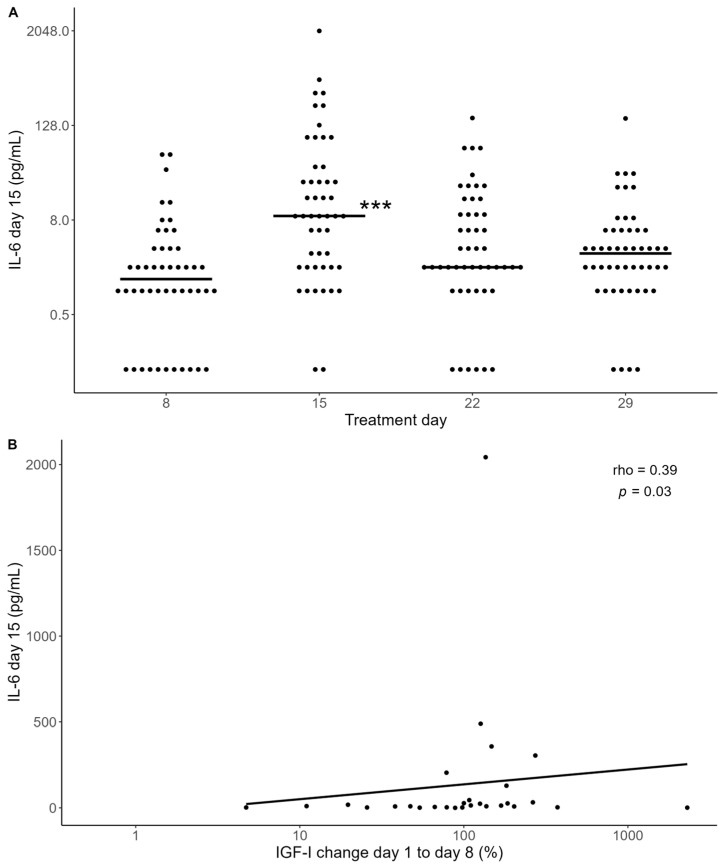
(**A**) IL-6 measurements for a subgroup of 49 patients. Horizontal black lines indicate medians. Il-6 levels were significantly increased on day 15 compared with day 8 (*** *p* < 0.001). Statistical evaluation using mixed-model analysis. (**B**) Association between IL-6 levels on day 15 and relative change in IGF-I levels from day 1 to 8. Statistics by Spearman correlation (rho = 0.39, *p* = 0.03). The black line represents a linear regression line fitted to the data.

**Figure 4 ijms-25-09582-f004:**
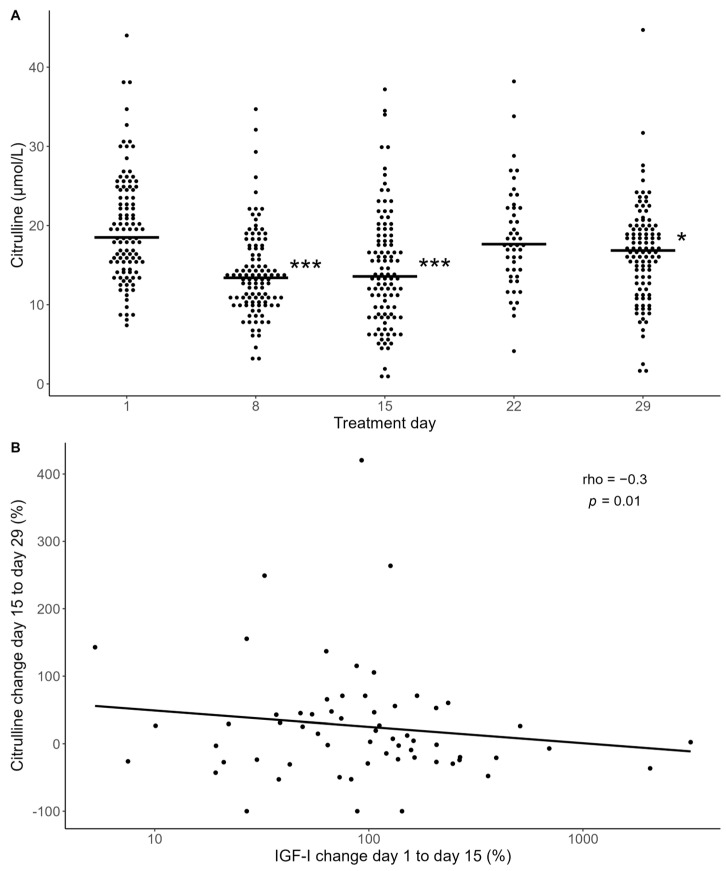
(**A**) Plasma citrulline levels during induction therapy. Asterisk indicates where citrulline levels are significantly lower than day 1 levels by a mixed-model analysis. * *p* < 0.05, *** *p* < 0.001. (**B**) Association between relative change in citrulline levels from day 15 to day 29 and relative change in IGF-I levels from day 1 to 15. Statistics by Spearman correlation analysis (rho = −0.3, *p* = 0.01). The black line represents a linear regression line fitted to the data.

**Figure 5 ijms-25-09582-f005:**
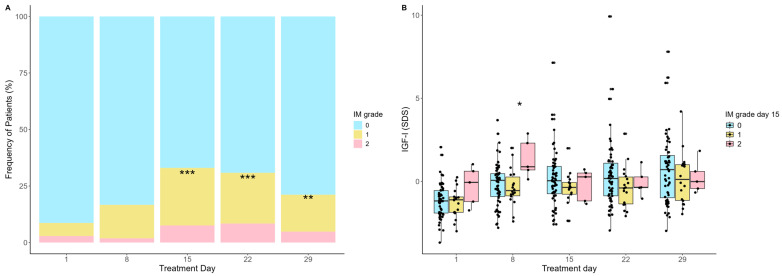
(**A**) Clinical intestinal mucositis (IM) severity during induction treatment. The incidence of IM was significantly higher on days 15, 22, and 29 than day 1, *** *p* < 0.001, ** *p* < 0.01. (**B**) IGF-I levels during the induction treatment according to severity of IM on day 15 (peak day). IGF-I levels on day 8 were significantly higher in patients with IM grade II on day 15 compared with patients with grade 0-I, * *p* = 0.02. Statistics by Mann–Whitney U-test.

**Table 1 ijms-25-09582-t001:** Patients characteristics (N = 114).

**Sex**	
Male	67 (59%)
Female	47 (41%)
**Age**	
Median (range), years	4.2 (1.17)
**ALL immunophenotype**	
Pre-B-cell ALL	99 (86.8%)
T-cell ALL	12 (10.5%)
Biphenotypical	3 (2.6%)
**Induction risk group ^1^**	
Non-High risk	98 (86%)
High risk	16 (14%)
**Final risk group ^2^**	
Standard risk	48 (42%)
Intermediate risk	56 (49%)
High risk	7 (6%)
Other	3 (3%)

^1^ Risk stratification 1 on treatment day 1: At diagnosis, patients were assigned to high risk (T-ALL and/or WBC ≥ 100 × 10^9^/L) or non-high risk (all other patients) [34]. ^2^ Risk stratification 2 on treatment day 29 [34]. ALL, acute lymphoblastic leukemia.

## Data Availability

The data presented in this study are available from the corresponding author, M.E.S., upon reasonable request.

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
