# Peer review of "Upregulation of Insulin-like Growth Factor-I in Response to Chemotherapy in Children with Acute Lymphoblastic Leukemia"

_ijms, 2024, doi:10.3390/ijms25179582_

Round 1

Reviewer 1 Report (Previous Reviewer 2)

Comments and Suggestions for Authors

Thank you for your replies to my comments.

Author Response

Thank you very much for taking the time to review this manuscript. 

Reviewer 2 Report (New Reviewer)

Comments and Suggestions for Authors

General Comments:  This is an interesting study in which the authors showed reduced IGF-I concentrations in children with newly diagnosed acute lymphoblastic leukemia (ALL) but increased concentrations in response to chemotherapy, with the intensity of the response reflecting the severity of toxicity. The finding of increased IGF-I concentration preceding the observed increases in established inflammatory markers suggests its potential as an early biomarker for chemotherapy-induced tissue injury and inflammation.  I have no concerns with the design of the experiment, the statistical analysis of the data, or the results and conclusions.  The manuscript is well-written and requires only minor editing.

Specific Comments:

ABSTRACT:  The Abstract is normally written as a single paragraph.

Line 25 and elsewhere:  I was taught it should be “IGF-I concentrations” instead of “IGF-I levels”.

Line 33:  Change “IL-6(rho” to “IL-6 (rho”. (Leave a space).

Line 39:  Delete “the”.

Line 84:  Change to “…are outlined in detail in Table 1”.

Table 1:  In some categories, the percentages do not sum to 100%.  Why is that?  Include the units of measurement (years) for the age variable.

Figure 1:  This figure is difficult to read.  Please try to improve the resolution.

Figure 4:  P = 0.01 or P < 0.01?

Line 239:  Correct the spelling of “asterisk”.

Lines 310-312:  Avoid single-sentence paragraphs.

Lines 348-349: “detail” instead of “details”.

Line 360:  Change to “…were converted to sex- and…”.

Lines 366-367:  I am sorry, but I do not understand the meaning of “plasma CRP was measured on a median of 19 out of 29 (range 4-29) for each patient”.  19 out of 29 days?  What are the units of measure for these numbers?

Lines 374-376:  Avoid single-sentence paragraphs.

Line 375:  Delete “by”.  “by using” is redundant.

Line 386:  Change to “treatment center”.

Line 388:  Does “IGF-I and IGFBP-3 levels (due to its skewness)” mean that IGF-I and IGFBP3 were both skewed or only IGFBP3 was skewed?

Lines 389-390:  This is not a complete sentence.  Please reword.

Comments on the Quality of English Language

The manuscript is well-written and requires only minor editing.

Author Response

Reviewer 3 Report (New Reviewer)

Comments and Suggestions for Authors

This manuscript studied the level of Insulin-like Growth Factor-I in response to chemotherapy in 114 children (age 1-17 years old) with acute lymphoblastic leukemia . The results showed rapid increase in plasma levels of IGF-I and IGFBP-3 after initiation of chemotherapy treatment, correlating with the severity of tissue toxicity in terms of an increased systemic inflammatory response and a delayed recovery of intestinal epithelial cells. However, I have some comments and suggestions.

Line 19: Please rewrite the abstract to be one paragraph according to the journal guidelines "The abstract should be a single paragraph and should follow the style of structured abstracts, but without headings".

Line 25: please verify this abbreviation "NOPHO".

Line 36: please rewrite these results to be more focused and specific.

Line 47: The reference [1] was published since 2015, 9 years ago. Please recheck again the updated literature to confirm if the percent has changed or not.

Line 78: Please revise this paragraph to be more specific and briefly mention why this work is important and highlight the main conclusions that are not mentioned.

Line 85: Rewrite the title of table as not need to be bold, the same as in figures along the manuscript.

Line 90: please add more details about why did you use "Median" of age at table 1.  However, the age ranged from 1 to 17 years old at which the immune system showing a developmental differences during this time of age.

Line 111: Please verify why the number of samples at Day 1 is 84 not 114 as your total treated patients.

Line 195: Please revise the label of vertical line at figure 3 A.

Line 250: Please add more explanation for the criteria of these 3 grades to show the difference between grade 0, I and II.

Line 257: I suggest using 3 different colors for the grades (0-II) to show how big and or  how many patients in each grade during these different times of treatments including grade 0.

Line 258: The dark grey color used for grade 1 intestinal mucositis grade and showed above the grade 2 (light grey). Please revise and choose which one should be shown first at the bottom of column.

Line 267: Please rewrite the discussion it uses a separate point of your findings to discuss with others. The findings and their implications should be discussed in the broadest context possible, and the limitations of the work highlighted.

Line 268: the results showing until day 29 of ALL therapy means like 4 weeks not 6 weeks. Please revise.

Line 276: please review the below lines 288-292 in which you explained and discussed "the mechanism underlying your findings".

Line 288: please mention the references that you mean at " together with our study".

Line 318: I suggest highlighting this paragraph as a separate "conclusion" section with bold title.

Line 325: Please add more benefits for your findings to use a recommendation for the future treatment of children with this disease "ALL".

Line 329: Why the age here is 1-18? However, the age mentioned before is (1-17), please revise and verify.

Line 333: This study ended at 2018 in the past 6 years ago. Is that any update about the treatment protocol NOPHO during 2018 until NOW? Is they used a new medication that may affect your results.

Line 371: " on days 8, 370 15, 22, and 29", However the figure 3A shows extra results for day 30. Please verify if you did it and be consistent.

Line 378: Please verify that the abdominal pain and diarrhea caused by intestinal mucositis not from other abdominal organs problem.

References: number 8, 28, 36, 41, 43 and 53 are incomplete. Please revise.

Comments on the Quality of English Language

 Minor editing of English language required.

Round 2

Reviewer 3 Report (New Reviewer)

Comments and Suggestions for Authors

The suggested corrections looks good and the manuscript improved more than before. I noticed that you have 60 references not 58 like before, thanks for that!. But I have a comment please. References number 1, 40, 42, 48, 49 and 57 are incomplete. Please revise and bold the year through all references list.

Comments on the Quality of English Language

Minor editing of English language required.

This manuscript is a resubmission of an earlier submission. The following is a list of the peer review reports and author responses from that submission.

Round 1

Reviewer 1 Report

Comments and Suggestions for Authors

In their manuscript the authors describe the results of a prospective study on a cohort of pediatric patients with acute lymphoblastic leukemia (ALL), investigating the correlation between plasma IGF-I levels and markers of intestinal damage and systemic inflammation during the induction treatment. They found that an increase of circulating IGF-1 levels in response to chemotherapy is related to the severity of intestinal damage and it occurs earlier than a rise in CRP and IL-6. The study is well conducted and the statistics is  adequately performed. However, information about the degrees of oral and gastrointestinal mucositis experienced in the analyzed cohort and their correlation with IGF-1 levels is missing. Also, to increase the relevance of this work we suggest to include a brief paragraph in the Discussion section highlighting the potential clinical use of IGF-1 as a predictive biomarker of gastrointestinal toxicity during ALL induction chemotherapy in terms of prophylactic measures.

Reviewer 2 Report

Comments and Suggestions for Authors

Dear authors,

Here are my suggestions for improvement in the manuscript:

Line 24: “In this prospective study, we included 114 patients with newly diagnosed ALL». More details about the characteristics of patients and for how long you collected those patients. I saw that you have details in the materials and methods.

Line 29: «At ALL diagnosis, IGF-I levels were decreased compared to healthy controls (median (quartiles): -29 1.2 standard deviation score (SDS)». Which is the healthy controls? I think you should correct “is” to “are” healthy control. I think in the abstract just a very short mention.

Line 42: The keyword “gut microbiota” what for? I did not find a connection with the abstract. Someone must read the whole paper to understand that.

Line 52-52: «In this prospective study, we included 114 patients with newly diagnosed ALL» Please be more specific. You explain in Materials and Methods.

Line 74: “These findings suggest that inflammation, IGF-I production, and tissue healing are functionally interrelated” Where are comments about the intestinal epithelial?

Line 75: “Dynamic relations between IGF-I  and markers of intestinal damage and systemic inflammation” You continue with intestinal damage and systemic inflammation. Where is now the tissue healing?

I think you should connect more clearly the two specific comments.

Line 79-80: “A total of 114 pediatric patients with ALL were included in the study. Patient characteristics are outlined in Table 1». Ι believed you should describe briefly the characteristics of pediatric patients, not only the table. The levels of glucose?

Line 95-96: “IGF-I and IGFBP-3 levels were not associated with ALL immunophenotype or risk classification groups at any time point”. You should present, at least, some results, not only sttement. You are in the section of results. Or you could write (data not shown).

Line 97-99: “IGF-I and IGFBP-3 levels were not associated with ALL immunophenotype or risk classification groups at any time point”. I think these lines could be on the next page in paragraph 2 of the results. Because on the same page seems results for IGF-1 and IGFBP-3.

Line 241-242: “This suggests that a larger increase in IGF-I is associated with a slower recovery rate of citrulline”. A discussion also with IGFBR-3. A discussion also with healthy people.

Line 254: “Including co-morbidities and nutritional status” You could refer to some examples here.

Line 280-281: “Interestingly, our studies suggest that IGF-I is rapidly released in response to acute tissue damage, with the magnitude of its response adjusted to the severity of the initiating trauma”. I think at that point you should explain according to your results adjusted to the severity of the initiating trauma?.

Line 288-289: “However, the study was limited by the lack of power to further establish IGF-I as a biomarker for the prediction of the overall clinical outcomes”. I believe that was very high goal for the study to be biomarker the IGF-I. You should explain that you give some new data that IGF-I could be used in the future as a biomarker.

Line 390: “oktober” you should spell and write correctly in english as also lines 393, 414 and so on in all references.

Sincerely